# Fault Diagnosis of Rolling Bearing Based on a Priority Elimination Method

**DOI:** 10.3390/s23042320

**Published:** 2023-02-19

**Authors:** Chuan Xiang, Jiahui Zhou, Bing Han, Weichen Li, Hongge Zhao

**Affiliations:** 1College of Marine Electrical Engineering, Dalian Maritime University, Dalian 116026, China; 2National Engineering Research Center of Ship & Shipping Control System, Shanghai Ship and Shipping Research Institute Co., Ltd., Shanghai 200135, China

**Keywords:** priority elimination, fault diagnosis, rolling bearing, SSAE, XGBoost

## Abstract

Aiming at the fault diagnosis accuracy of rolling bearings is not high enough, and unknown faults cannot be correctly identified. A priority elimination (PE) method is proposed in this paper. First, the priority diagnosis sequence of faults was determined by comparing the ratios of the inter-class distance to the intra-class distance for all faults. Then, the model training and fault diagnosis were carried out in order of the priority sequence, and the samples of the fault that had been identified were eliminated from the data set until all faults were diagnosed. For the diagnosis model, the stacked sparse auto-encoder network (SSAE) was selected to extract the features of the vibration signal. The extreme gradient boosting algorithm (XGBoost) was chosen to identify the fault type. Finally, the method was tested and verified by experimental data and compared with classical algorithms. Research results indicate the following: (1) with the addition of PE based on SSAE-XGBoost, the fault diagnosis accuracy can be improved from 96.3% to 99.27%, which is higher than other methods; (2) for the test set with the samples of unknown faults, the diagnosis accuracy of SSAE-XGBoost with PE can reach 92.34%, which is nearly 6% higher than that without PE and is also obviously higher than other classical fault diagnosis methods with or without PE. The PE method can not only improve the diagnosis accuracy of faults but also identify unknown faults, which provides a new method and way for fault diagnosis.

## 1. Introduction

As an essential component of rotating machinery, the rolling bearing is widely used in automobile manufacturing, aerospace, numerical control machines, and various electromechanical equipment [1]. Once the bearing breaks down, it may cause damage to the equipment and even cause significant economic losses and casualties. According to the research by Henriquez P et al. [2], bearings are the most easily damaged components in electromechanical equipment, whose failure rate accounts for 41% of total failures. Therefore, it is of great significance to study fault diagnosis methods of rolling bearings to ensure the stable and reliable operation of electromechanical equipment [3].

Currently, the methods commonly used for bearing fault diagnosis can be divided into two stages: feature extraction and fault identification [4].

In terms of feature extraction, the traditional method is to use the signal processing method to extract the fault features from the vibration signal in the time domain, frequency domain, or time–frequency domain [5]. It mainly includes wavelet transform (WT) [6], empirical mode decomposition (EMD) [7], short-time Fourier transform (STFT) [8], etc. In recent years, deep learning theory has become more and more mature, and computers’ computing power has been significantly improved. Many deep learnings, such as the convolutional neural network (CNN), the deep belief network (DBN), the auto-encoder (AE), and the long short-term memory network (LSTM) [9], etc., have been applied in fault diagnosis due to the fact that they can deal with complex and high-dimensional problems in massive data that cannot be solved by shallow learning [10], and they have many advantages including high efficiency, plasticity, and universality. CNN is a multi-layer neural network model in supervised learning, mainly composed of convolutional, pooling, and fully connected layers. It realizes fault diagnosis by extracting local features of the vibration signal layer by layer. Hoang et al. [11] proposed a method for bearing fault diagnosis based on the deep structure of CNN. A direct connection based on the CNN (DC-CNN) method was studied by Kim [12]. It can significantly improve training efficiency and diagnosis performance. Additionally, a rolling-element bearing fault diagnosis method using an improved 2D LeNet-5 network has been proposed to satisfy the requirements of fault diagnosis of rolling bearings [13]. DBN is composed of multi-layer restricted Boltzmann machines (RBM) and a layer of back-propagation (BP) neural networks [14]. An adaptive DBN model of fault diagnosis based on the Nesterov moment (NM) optimization was researched to extract deep representative features [15]. Gao et al. [16] proposed a new optimized adaptive DBN with high diagnostic accuracy and good convergence to analyze the vibration signal of rolling bearings. AE is an unsupervised learning method consisting of three layers of neurons. It has been widely used in the fault diagnosis of equipment. Shao et al. [17] proposed a novel fault diagnosis method for rolling bearings based on the deep wavelet auto-encoder (DWAE) and extreme learning machine (ELM). An intelligent fault diagnosis method of rotating machinery based on a semi-supervised deep sparse auto-encoder (SSDSAE) was presented by Zhao et al. [18]. Huang et al. [19] developed an innovative deep learning-based model, namely, memory residual regression auto-encoder (MRRAE), to improve the accuracy of anomaly detection in bearing condition monitoring recently.

In the aspect of fault identification, commonly used methods are the support vector machine (SVM), artificial neural network (ANN), ensemble learning (EL), etc. SVM is a machine learning method based on statistical learning theory. It has been widely applied due to its high accuracy and good generalization ability [20]. Zhu et al. [21] input the fault feature vectors into an SVM classifier to automatically accomplish bearing fault identification. The structure of ANNs is often determined empirically, and their recognition accuracy is related to the number of training samples. BP is the most commonly used algorithm in ANN. Song et al. [22] improved the traditional BP neural network and increased the diagnosis efficiency of BP neural networks. To learn and distinguish features adaptively from the original data, a multiscale local feature learning method based on the BP neural network (BPNN) for rolling bearings’ fault diagnosis was proposed by J. Li [23]. Extreme gradient boosting (XGBoost), an ensemble learning method, has been proven to have high accuracy and fast processing time [24]. In reference [25], the XGBoost was adopted as the final classifier, and good results were achieved.

The fault diagnosis methods of rolling bearings described above can identify all fault types in the test set, but all the fault types they identified were trained and labeled in advance. If there are new unknown faults in the test set, these methods will identify them as the most similar faults and fail to identify them as belonging to new fault types. It will not only reduce the accuracy of fault diagnosis but may also cause serious harm to the electromechanical equipment.

Considering the above, the purpose of this paper is to present a novel priority elimination (PE) method combined with the stacked sparse auto-encoder network (SSAE) and the extreme gradient boosting algorithm (XGBoost) for the fault diagnosis of rolling bearings, to improve the diagnosis accuracy and correctly identify new unknown faults. The PE is used to determine the diagnosis sequence, SSAE is applied to extract fault features, and XGBoost is used to identify fault types. This paper is organized as follows. Section 2 expounds on the theoretical methods of the SSAE, XGBoost, and PE. The diagnosis procedure based on PE is described in detail in Section 3. Section 4 verifies the effectiveness of the proposed method using the experimental data of rolling bearing faults of Case Western Reserve University as an example. Conclusions are enclosed in Section 5.

## 2. Related Methods

### 2.1. Priority Elimination Method

The PE method is mainly used to determine the diagnosis sequence of different fault types. Its operation process is described as follows:

Step 1. Adopt t-distributed stochastic neighbor embedding (t-SNE) [26] to reduce vibration signal features from multi-dimension to two-dimension.

Step 2. Calculate the intra-class distance between different samples of each fault type to form an intra-class distance matrix *S_w_*.

Suppose $G_{p}$ is a fault type in the training set, which contains $n_{p}$ samples. Then, its intra-class distance D is
(1)$$D=\frac{1}{n_{p}^{2}}\overset{n_{p}}{\underset{k=1}{\sum }}\overset{n_{p}}{\underset{l=1}{\sum }}d\left(X_{k}^{\left(i\right)},X_{l}^{\left(i\right)}\right)$$
where $X_{k}^{\left(i\right)}$ and $\;X_{l}^{\left(i\right)}$ are different samples in $G_{p}$, and d() is the Euclidean distance between different samples.

The calculated intra-class distances of all fault types are taken as the diagonal elements to form the intra-class distance matrix *S_w_*.

Step 3. Calculate the inter-class distance between different samples in different fault types to form an inter-class distance matrix *S_b_*.

Assume that $G_{p}$ and $G_{q}$ are two different fault types in the training set, which contains $n_{p}$ and $n_{q}$ samples, respectively. The distance between these two fault types can be expressed in various ways, such as the nearest distance method, which is defined as
(2)$$D_{\min}=\underset{i,j}{\min}\left|d_{ij}\right|$$
where $d_{ij}$ denotes the distance between the sample $X_{i}$ in $G_{p}$ and sample $X_{j}$ in $G_{q}$. The nearest distance method defines the shortest distance between two fault types as the inter-class distance.

On the contrary, if the maximum distance between two fault types is defined as the inter-class distance, it is called the farthest distance method.
(3)$$D_{\max}=\underset{i,j}{\max}\left|d_{ij}\right|$$

In addition, there is an intermediate distance method, which is a compromise between the nearest and the farthest distance methods. It combines $G_{p}$ and $G_{q}$ to form a new type $G_{n}$, and then calculates the distance between other types $G_{l}$ and $G_{n}$. This distance is called the intermediate distance, which is defined as
(4)$$D_{nl}^{2}=\frac{1}{2}D_{lp}^{2}+\frac{1}{2}D_{lq}^{2}-\frac{1}{4}D_{pq}^{2}$$

If the intermediate distance method considers the number of samples in each fault, it is called the barycenter distance method, which is defined as
(5)$$D_{nl}^{2}=\frac{n_{p}}{n_{p}+n_{q}}D_{lp}^{2}+\frac{n_{p}}{n_{p}+n_{q}}D_{lq}^{2}-\frac{n_{p}n_{q}}{{\left(n_{p}+n_{q}\right)}^{2}}D_{pq}^{2}$$

This paper adopts the average distance method. The inter-class distance is the average distance between any two different samples in any two different fault types, which is defined as
(6)$$D_{pq}=\frac{1}{n_{p}n_{q}}\underset{x_{i}\in G_{p}}{\sum }\underset{x_{j}\in G_{q}}{\sum }d_{ij}$$

According to Formula (6), the inter-class distance between each fault and other faults is calculated, and then the sum of the inter-class distance between each fault type and other fault types is calculated to form the inter-class distance matrix *S_b_*.

Step 4. Calculate the ratio of the inter-class distance to the intra-class distance for each fault. The larger the inter-class distance between different fault types, while the smaller the intra-class distance of the same fault, the larger the ratio of them. This means that this fault has more obvious features and is easier to identify. The calculated distance ratios are ranked in descending order, and the priority diagnosis sequence of faults is obtained.

The operation process described above can be summarized in a flow chart, as shown in Figure 1.

### 2.2. SSAE Network

The block diagram and unfolding structure of the auto-encoder network (AE) are shown in Figure 2.

AE consists of an encoder and a decoder. x, y, and $\tilde{x}$ correspond to the input layer, hidden layer, and output layer, respectively. Its working process is that the encoder transforms the input vector x into the coding vector y, and then the decoder converts y into the output vector $\tilde{x}$. $\tilde{x}$ is also known as the reconstruction vector of x. The transformation form is as follows:(7)$$y=f\left(x\right)$$
(8)$$\tilde{x}=g\left(y\right)=g\left(f\left(x\right)\right)$$
where f and g are the activation functions for the encoding and decoding processes, respectively.

AE usually selects the sigmoid activation function, whose expression is
(9)$$sigmoid\left(x\right)=\frac{1}{1+e^{-x}}$$

AE does not focus on the network’s output, but on the coding, i.e., the mapping from input to output. The coding vector y is a mapping of the input vector x. The proximity between the output vector $\tilde{x}$ and the input vector x is calculated to measure the quality of the AE network construction. The mean square error is used as the loss function L, which is also called the reconstruction error:(10)$$L\left(W,b\right)=\frac{1}{d}\overset{d}{\underset{i=1}{\sum }}\left(\frac{1}{2}\|x_{i}-{\tilde{x}}_{i}\|{}^{2}\right)$$
where $\left(W,b\right)$ is the parameter set of the network, W is the weight matrix, b is the offset vector, and d is the number of samples.

The loss function is minimized by iterative optimization. At this time, it is considered that the network already contains most of the information about the input vectors, and the parameter set has obtained the best implicit relationship for the input vectors.

The sparse auto-encoder (SAE) is obtained by adding constraints to the AE. To avoid the overfitting of the network, a sparse penalty term is added to the original loss function. The Kullback–Leibler divergence is generally selected by SAE as the sparse penalty term of the network, and the improved loss function J is
(11)$$J\left(W,b\right)=L\left(W,b\right)+\alpha \overset{k}{\underset{m=1}{\sum }}KL\left(\rho \|\rho_{m}\right)$$
(12)$$KL\left(\rho \|\rho_{m}\right)=\rho \ln\left(\frac{\rho }{\rho_{m}}\right)+\left(1-\rho \right)\ln\left(\frac{1-\rho }{1-\rho_{m}}\right)$$
where α is the coefficient of the sparse penalty term, ρ is the sparse parameter, and $\rho_{m}$ is the average activation of the m-th node of the hidden layer.

Ordinary SAE only has three layers and thus has difficulty learning all the interior features of the input vector and obtaining the deep hidden relationship of the data. Therefore, BENGIO Y et al. proposed an SAE network [27] which forms a stacked SAE network (SSAE) by stacking the shallow SAEs. In SSAE, each SAE is trained separately to obtain the parameters of each layer of the network, and the hidden layer of the lower SAE is used as the input layer of the higher SAE. Due to the increased network depth, SSAE is prone to overfitting, so it is necessary to add a regularization term to the improved loss function J. The new loss function $J_{\mathrm{SSAE}}$ is expressed as
(13)$$J_{\mathrm{SSAE}}=J\left(W,b\right)+\frac{\lambda }{2}\overset{n_{l}-1}{\underset{l=1}{\sum }}\overset{s_{l}}{\underset{i=1}{\sum }}\overset{s_{l+1}}{\underset{j=1}{\sum }}{\left(W_{ji}^{\left(l\right)}\right)}^{2}$$
where λ is the coefficient of the control regularization term, $n_{l}$ is the total number of network layers, $s_{l}$ is the number of nodes in layer l, and $W_{ji}^{\left(l\right)}$ is the network parameter matrix of layer l. The coding process can be expressed as
(14)$$y^{\left(t\right)}=f\left(x^{\left(t\right)}\right),\forall t=1,2,\cdots ,n_{l}$$
(15)$${\tilde{x}}^{\left(t\right)}=W^{\left(t\right)}y^{\left(t-1\right)}+b^{\left(t\right)}$$

Assume there are n coding layers in the coding process. The decoding process is then expressed as
(16)$$y^{\left(t+n\right)}=f\left(x^{\left(t+n\right)}\right),\forall t=1,2,\cdots ,n_{l}$$
(17)$${\tilde{x}}^{\left(t+n+1\right)}=W^{\left(n-t\right)}y^{\left(n+t\right)}+b^{\left(n-t\right)}$$

### 2.3. XGBoost Algorithm

XGBoost is an algorithm based on a decision tree proposed by Dr. Chen at the University of Washington [28]. The objective function is defined as
(18)$$Obj^{\left(t\right)}=\overset{n}{\underset{i=1}{\sum }}l\left(y_{i}^{’},{\hat{y}}_{i}^{’\left(t-1\right)}+f_{t}\left(x_{i}^{’}\right)\right)+\Omega \left(f_{t}\right)$$
where $y_{i}^{’}$ is the actual value, ${\hat{y}}_{i}^{’\left(t-1\right)}$ is the predicted value for round t−1, $f_{t}\left(x_{i}^{’}\right)$ is the score function of samples in round t, and the final predicted value is the sum of them. $\Omega \left(f_{k}\right)$ represents the complex function of the tree; the smaller its value, the lower the tree’s complexity and the stronger the generalization ability.
(19)$$\Omega \left(f\right)=\gamma T+\frac{1}{2}\lambda \|\omega \|{}^{2}$$
where T is the number of leaf nodes, ω represents the value or class of the node, and λ and γ are scale factors. $\|\omega \|{}^{2}$ represents $L_{2}$ regularization of ω.

Next, the loss function $J_{\mathrm{SSAE}}$ described by Equation (19) is expanded by the second-order Taylor expansion, and then the first- and second-order derivatives are obtained. Finally, the objective function $Obj^{\left(t\right)}\;$ can be obtained after sorting:(20)$$Obj^{\left(t\right)}=-\frac{1}{2}\overset{T}{\underset{j=1}{\sum }}\frac{G_{j}^{2}}{H_{j}+\lambda }+\gamma T$$
where $G_{j}$ and $H_{j}$ are the sums of the first- and second-order derivatives, respectively.

## 3. Fault Diagnosis Process

First, the PE method is adopted to prioritize the fault diagnosis sequence, and then the SSAE-XGBoost model is used to extract fault features and identify fault types. The whole fault diagnosis process is shown in Figure 3.

As can be seen from Figure 3, the detailed diagnosis steps are as follows:

Step 1. Divide the vibration signal of the rolling bearing into a training set and a test set.

Step 2. Determine the priority diagnosis sequence of faults according to the PE method, and assume the priority diagnosis sequence is *X*_1_ > *X*_2_ > *X*_3_…> *X_n_*.

Step 3. Train the SSAE-XGBoost model according to the priority diagnosis sequence of faults above. The detailed training steps are as follows: First, all the data in the training set are used for model training to diagnose the fault *X*_1_ with the highest diagnostic priority and obtain the first diagnosis model. Then, the samples of *X*_1_ fault are eliminated from the training set. Next, the remaining samples are used for model training to diagnose the fault *X*_2_ and obtain the second diagnosis model. Then, the samples of *X*_2_ fault are eliminated from the training set…Repeat the above steps until all the trained SSAE-XGBoost models of fault diagnosis are obtained.

Step 4. Verify the PE-SSAE-XGBoost method by the test set. Perform the fault diagnosis in the priority diagnosis sequence until all known faults are diagnosed. If there are remaining samples in the test set, they are considered as samples of unknown new faults.

## 4. Experimental Validation

### 4.1. Experimental Data

In this study, the experimental data set of rolling bearings was from the Electronic Engineering Laboratory of Case Western Reserve University (CWRU). The experimental platform is shown in Figure 4.

It consists of a motor, a torque sensor, a power tester, and an electronic controller. The rolling bearing to be diagnosed drives the motor rotation. It is generally composed of an inner ring, an outer ring, a rolling element, and a cage. Its main faults are the inner ring fault (IRF), outer ring fault (ORF), and ball fault (BAF). The test bearing is a deep groove ball bearing from the SKF company (6205-2RSIEF). The artificial damage of the test bearing was formed by discharging and ablating at a single point. The damage points of ORF were set at 3 o’clock, 6 o’clock, and 12 o’clock on the outer ring clock. The vibration signals were measured by acceleration sensors mounted on the fan end (FE) and drive end (DE) of the motor and recorded by a 16-channel data recorder with a sampling frequency of either 12 kHz or 48 kHz.

The data selected for this study were FE and DE data with 3 hp, a sampling frequency of 12 kHz, and a rotating speed of 1730 rpm. They included the normal state and nine kinds of fault states. The nine fault states were BAF, IRF, and ORF data with 0.1778, 0.3556, and 0.5334 mm diameters of the damage point, respectively. For ORF, the fault data were selected with the damage point at 6 o’clock. There were 102,400 consecutive sampling points for each fault state and 204,800 sampling points for the normal state. The selected data are listed in Table 1. A total of 70% were used as the training set and the remaining 30% were used as the test set.

The computing platform is described as follows: the software used was PyCharm development software based on the Python environment. Main configuration parameters of the PC were CPU (Intel Core i7-8750H, Santa Clara, CA, USA), Graphics card (NVIDIA GeForce RTX 2060, Santa Clara, CA, USA), and Memory (16 GB).

### 4.2. Priority Diagnosis Sequence

According to the PE method, the intra-class distance *S_w_* and inter-class distance *S_b_* of each fault state described in Table 1 are calculated first. The calculated *S_w_* is illustrated in Figure 5, where fault states 1 (DE) and 1 (FE) represent the IRFs with damage diameter 0.01778 mm labeled 1 in Table 1, which are collected on DE and FE, respectively. The meanings of other fault states in Figure 5, and the following figures are similar. As shown in Figure 5, the *S_w_* of 8 (DE), i.e., the intra-class distance of BAF with the damage diameter 0.3556 mm at the drive end, is the farthest, and the *S_w_* of 2 (FE), i.e., the intra-class distance of IRF with the damage diameter 0.3556 mm at the fan end, is the closest. The calculated *S_b_* between different fault states is displayed in Figure 6.

Next, according to Figure 6, the sum of the inter-class distance *S_b_* between different fault states for each fault state is calculated, as shown in Figure 7. It can be seen that the inter-class distances of 1 (FE) and 2 (DE) are the farthest and closest, respectively.

Then, according to the calculated *S_w_* and *S_b,_* as shown in Figure 5 and Figure 7, the ratios of *S_b_* to *S_w_* for each fault state can be obtained, which are presented in Figure 8. As we can see from the ratios, the priority diagnosis sequence is 1 (FE) >2 (FE) >4 (DE) >…>8 (DE).

If the fault diagnosis is carried out according to the above priority sequence, the maximum diagnosis time is up to 71.58 s, which does not satisfy the requirement of fast and real-time fault diagnosis. Therefore, we consider all samples of the same fault type (IRF, BAF, or ORF) with different diameters as one data set for diagnosis, which can reduce the diagnosis time to 36.59 s. The average ratios of *S_b_* to *S_w_* for each fault type are calculated and summarized in Table 2.

It can be seen from Table 2 that the order of the ratios of *S_b_* to *S_w_* for all fault types, i.e., the diagnosis sequence, is IRF > ORF > BAF. A graphical diagram of the diagnosis process is illustrated in Figure 9. Due to the feature of the normal state (label 0) being obviously different from those of fault states (label 1~9), the fault diagnosis model will first diagnose the normal state, and then eliminate its samples from the data set. Next, faults are diagnosed according to the priority sequence of IRF, ORF, and BAF, and then samples of each fault type are eliminated from the data set in turn. If there are remaining undiagnosed samples in the data set after the last BAF fault has been diagnosed, they are identified as samples of unknown faults.

### 4.3. Diagnosis Results

In this part, the priority diagnostic sequences obtained by the PE method are put into SSAE and XGBoost to train the diagnostic models. Then, the effectiveness of the proposed method is verified by the test set.

The parameter setting of the SSAE-XGBoost model is listed in Table 3. The number of hidden layers of SSAE is set to 3, the network structure is set to 1024-512-256-128-10, the Adam algorithm is selected to optimize the network, and the number of iterations is set to 60. The most essential parameters of the classifier XGBoost are the maximum depth of a tree (max depth), the minimum sum of instance weight needed in a child (min child weight), the number of decision trees (n estimators), and the learning rate. We choose 5 as the maximum depth, the minimum sum of instance weight needed in a child is set to 1, 80 decision trees are constructed, and the learning rate is set to 0.12.

First, according to Figure 9, the normal state (label 0) in the test set is diagnosed, and all fault states are labeled as others. The obtained confusion matrix of the first diagnosis results is shown in Figure 10. It can be seen that the diagnosis accuracy reaches 98.2%; thus, the samples of normal state and fault states can be well distinguished. Then, the samples of the normal state (label 0) are eliminated from the test set.

Second, the IRFs with three different fault diameters (labels 1~3) are diagnosed, and remaining fault states are labeled as others. The confusion matrix of the second diagnosis results is shown in Figure 11. As we can see, the fault diagnosis accuracy is 99.67%. Then, the samples of all diagnosed IRFs are eliminated from the test set.

Third, the ORFs with three different fault diameters (labels 4~6) are diagnosed, and remaining fault states are labeled as others. The confusion matrix of the third diagnosis results is shown in Figure 12. It can be seen the accuracy of this fault diagnosis is 99.5%. Then, all diagnosed ORFs are eliminated from the test set.

Finally, the BAFs with three different fault diameters (labels 7~9) are diagnosed. The confusion matrix of the fourth diagnosis results is shown in Figure 13. As we can see, the fault diagnosis accuracy is 99%.

According to Figure 10, Figure 11, Figure 12 and Figure 13, the accuracy of fault diagnosis of rolling bearings with the PE method can be calculated as high as 99.27%. Without the PE method, the confusion matrix of the diagnosis results is shown in Figure 14, and the fault diagnosis accuracy is only 96.3%. The PE method significantly improves the diagnosis accuracy.

Furthermore, the proposed method is compared with classical fault diagnosis methods, such as CNN, SVM, and DBN, to further validate the advantages of this method. The comparison of diagnosis accuracy is listed in Table 4 (the SVM (94.78%) and DBN (93.19%) data are from previously published papers [29]). It can be seen from Table 4 that for CNN, SVM, and DBN, the PE method can improve the fault diagnosis accuracy. Especially for the proposed PE-SSAE-XGBoost, its fault diagnosis accuracy is significantly higher than other methods with or without PE.

The identification ability of this method for unknown faults is verified next. We select the samples of the IRF and BAF at 3 HP, 1730 rpm, and 0.7112 mm fault diameter and IRF at 0 HP, 1797 rpm, and 0.7112 mm fault diameter as the data set of the unknown faults and add them into the test set. The two-dimensional visualization of the new test set is shown in Figure 15.

After adding samples of unknown faults to the test set, the comparison results of fault diagnosis accuracy between the PE method and other methods are listed in Table 5.

It can be seen from Table 5 that for the new test set with unknown faults, the PE method can also improve the fault diagnosis accuracy of SSAE-XGBoost, CNN, SVM, and DBN. Especially, the fault diagnosis accuracy of PE-SSAE-XGBoost is up to 92.34%, whereas that of SSAE-XGBoost without the PE method is only 86.96%. The reason for the low diagnosis accuracy of these methods without PE is that they cannot distinguish unknown faults and identify them as the most similar known faults, which reduces the fault diagnosis accuracy.

## 5. Conclusions

This paper proposed a PE method which combines with SSAE and XGBoost to improve the fault diagnosis accuracy of rolling bearings and the identification ability of unknown faults. The following conclusions can be drawn:
(1)In terms of the improvement of the fault diagnosis accuracy, PE improves the fault diagnosis accuracy of all methods. The SSAE-XGBoost model combined with the PE method increases the fault diagnosis accuracy from 96.3% to 99.27%, which is also significantly higher than some classical algorithms with or without PE.(2)In the aspect of the identification of unknown faults, the fault data that do not appear in the training set are put into the test set. SSAE-XGBoost with PE can improve the accuracy of fault diagnosis from 86.96% to 92.34%, which is superior to other classical fault diagnosis methods with or without PE.

In conclusion, the proposed PE method has achieved good results in improving the accuracy of fault diagnosis and identifying unknown faults. It provides a new method for fault diagnosis, which is suitable for fault diagnosis of various mechanical equipment based on the data drive.

## Figures and Tables

**Figure 1 sensors-23-02320-f001:**
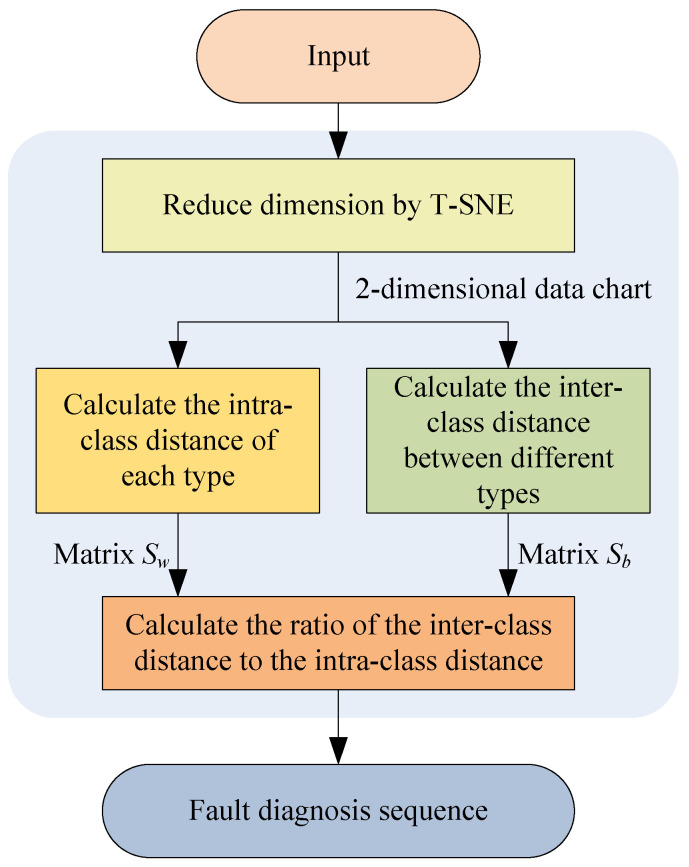
Operation process of the PE method.

**Figure 2 sensors-23-02320-f002:**
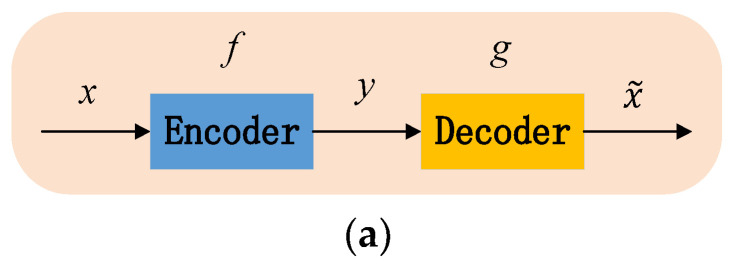
AE structure. (**a**) Block diagram. (**b**) Unfolding structure.

**Figure 3 sensors-23-02320-f003:**
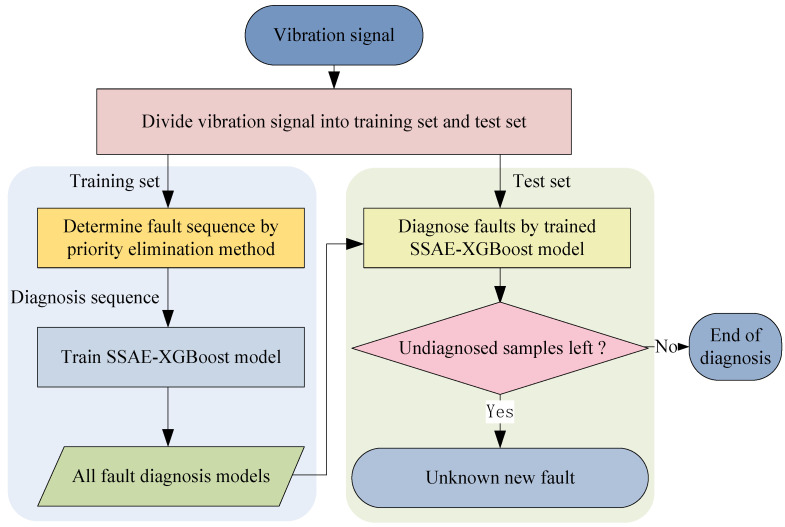
Flow chart of the priority elimination method.

**Figure 4 sensors-23-02320-f004:**
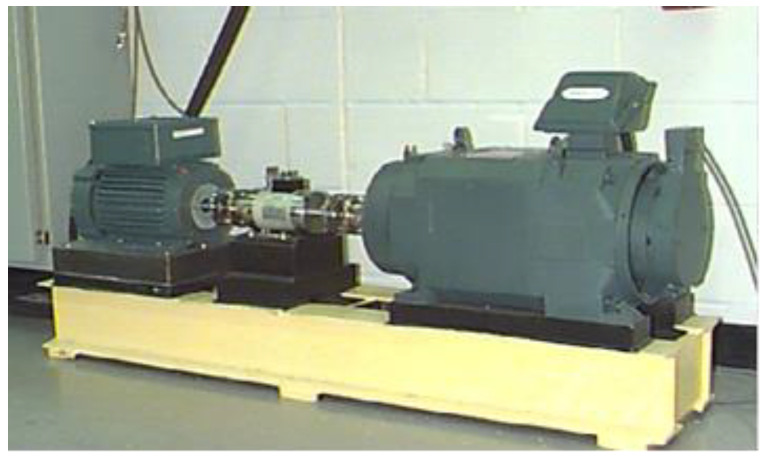
Experimental platform of rolling bearing.

**Figure 5 sensors-23-02320-f005:**
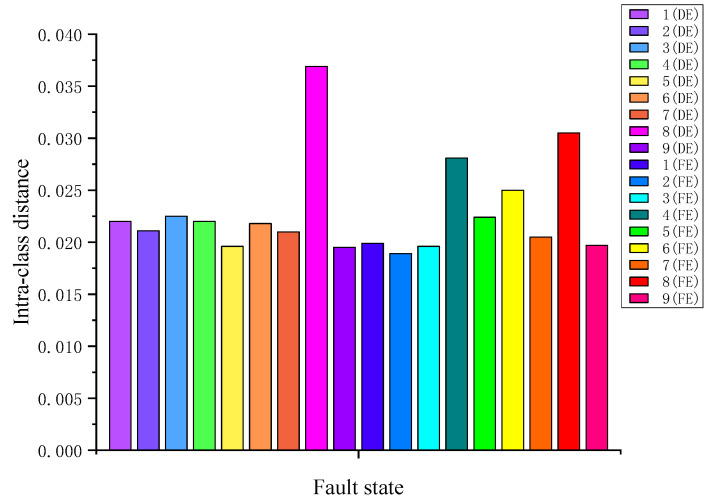
Intra-class distance *S_w_* of each fault state.

**Figure 6 sensors-23-02320-f006:**
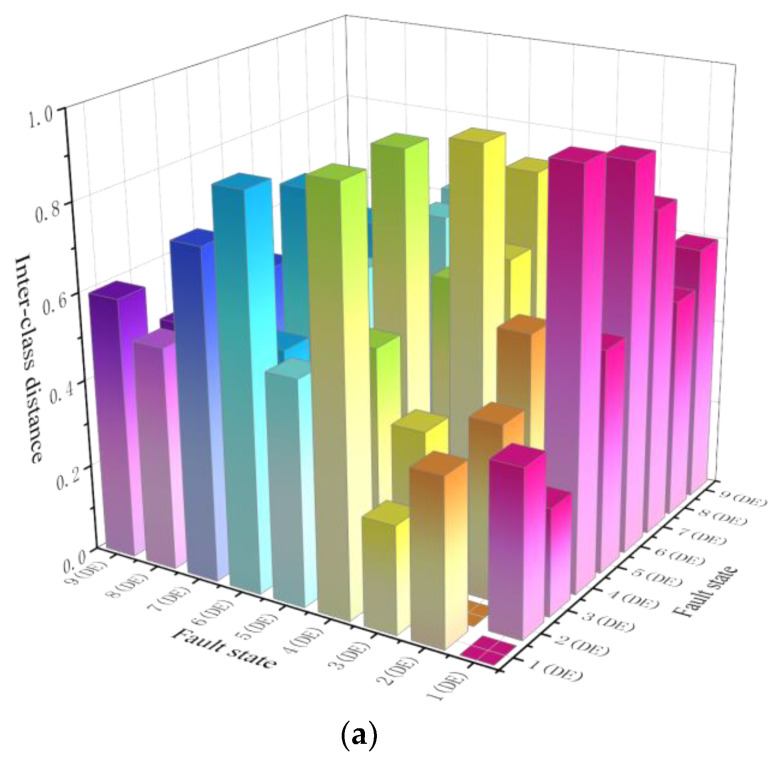
Inter-class distance *S_b_* between different fault states. (**a**) DE-DE inter-class distance *S_b_*. (**b**) DE-FE inter-class distance *S_b_*. (**c**) FE-FE inter-class distance *S_b_*.

**Figure 7 sensors-23-02320-f007:**
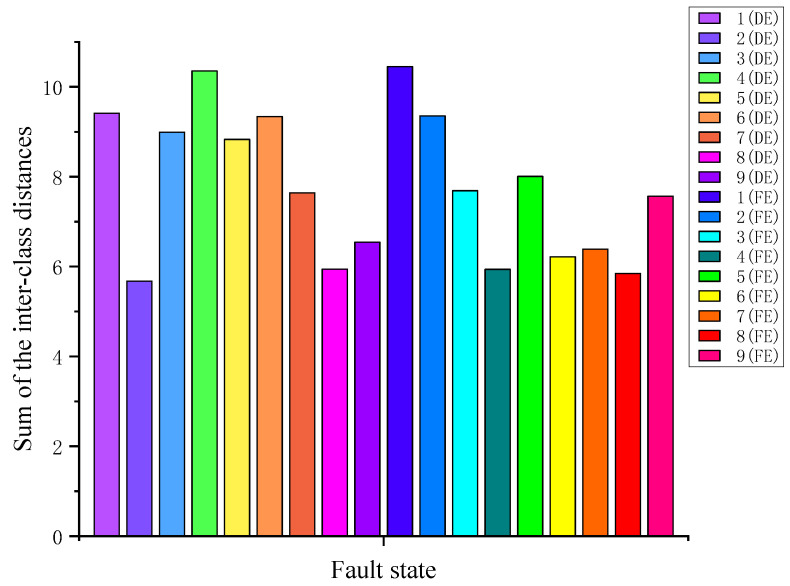
Sum of *S_b_* for each fault state.

**Figure 8 sensors-23-02320-f008:**
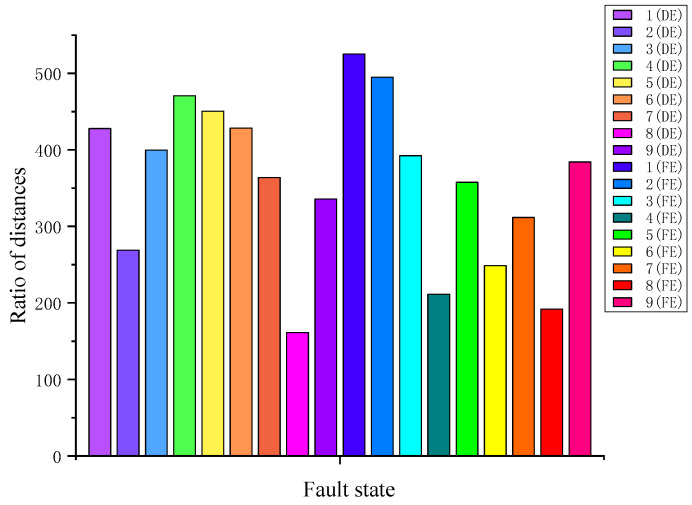
The ratios of *S_b_* to *S_w_* for each fault state.

**Figure 9 sensors-23-02320-f009:**
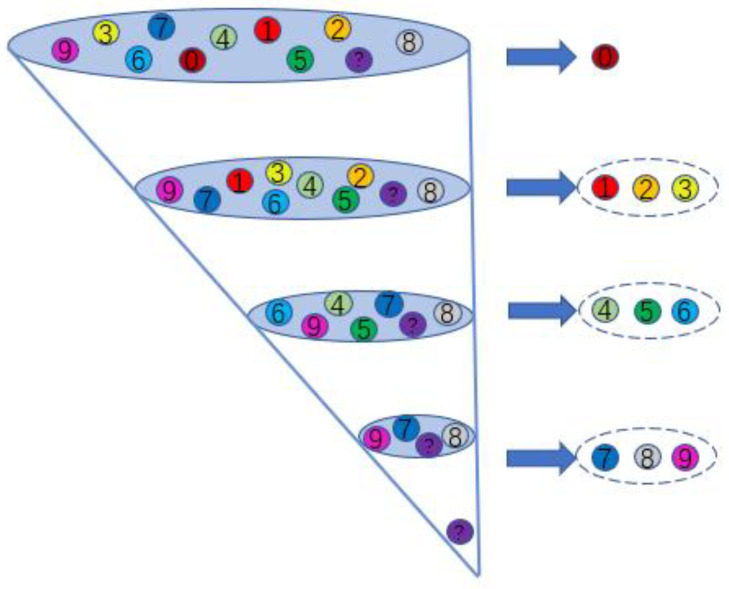
Graphical diagram of the diagnosis process.

**Figure 10 sensors-23-02320-f010:**
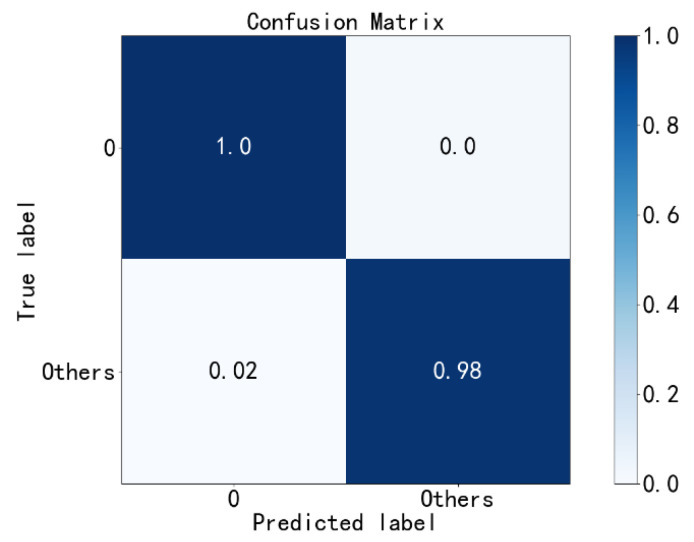
First fault diagnosis with the PE method.

**Figure 11 sensors-23-02320-f011:**
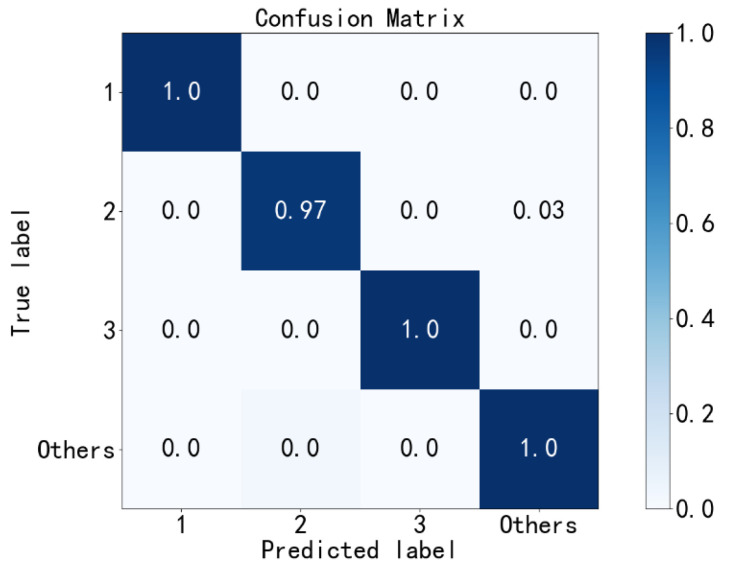
Second fault diagnosis with the PE method.

**Figure 12 sensors-23-02320-f012:**
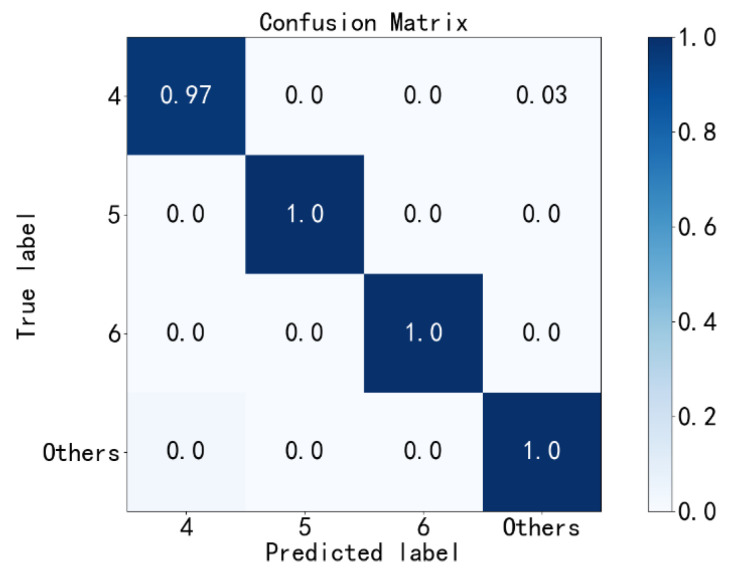
Third fault diagnosis with the PE method.

**Figure 13 sensors-23-02320-f013:**
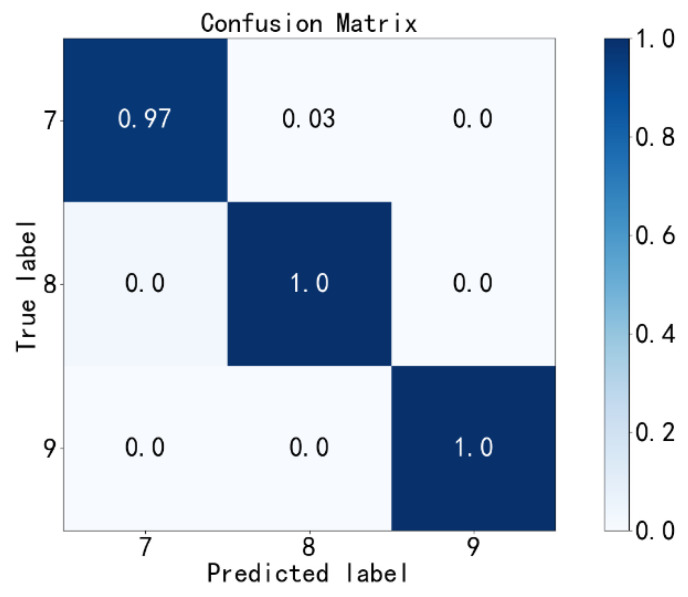
Fourth fault diagnosis with the PE method.

**Figure 14 sensors-23-02320-f014:**
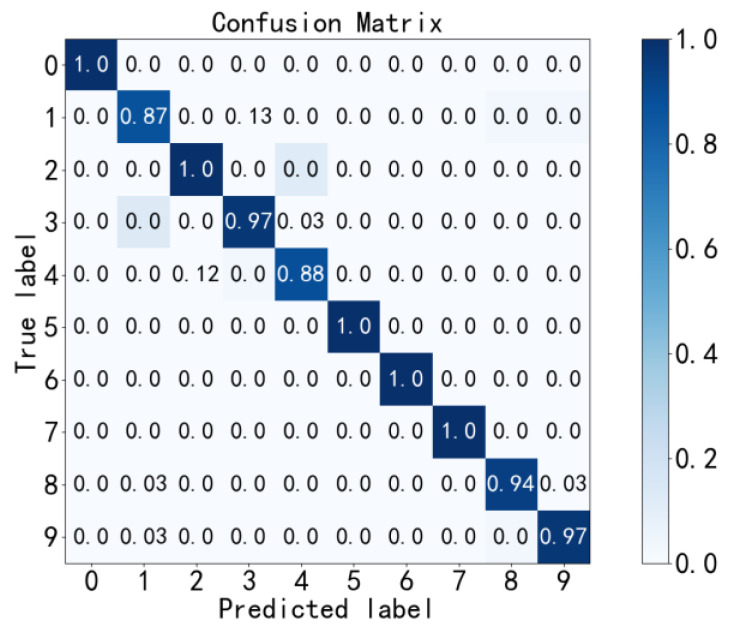
Fault diagnosis without the PE method.

**Figure 15 sensors-23-02320-f015:**
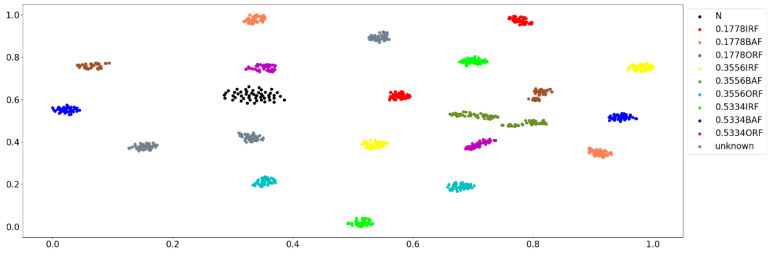
Two-dimensional visualization of the new test set.

**Table 1 sensors-23-02320-t001:** Experimental data of rolling bearings.

Fault Type	Fault Diameter/mm	Collecting Position	Sample Number	Label
IRF	0.1778	DE	102,400	1
		FE	102,400	1
	0.3556	DE	102,400	2
		FE	102,400	2
	0.5334	DE	102,400	3
		FE	102,400	3
ORF	0.1778	DE	102,400	4
		FE	102,400	4
	0.3556	DE	102,400	5
		FE	102,400	5
	0.5334	DE	102,400	6
		FE	102,400	6
BAF	0.1778	DE	102,400	7
		FE	102,400	7
	0.3556	DE	102,400	8
		FE	102,400	8
	0.5334	DE	102,400	9
		FE	102,400	9
Normal state	0	DE	204,800	0

**Table 2 sensors-23-02320-t002:** The ratios of *S_b_* to *S_w_* for each fault type.

Fault Type	IRF	BAF	ORF
Average ratio of *S_b_* to *S_w_*	418.17	291.37	361.22

**Table 3 sensors-23-02320-t003:** Parameter setting of SSAE-XGBoost.

SSAE	Number of hidden layers	3
	Network structure	1024-512-256-128-10
	Optimizer	Adam
	Iterations	60
XGBoost	Max depth	5
	Min child weight	1
	N estimators	80
	Min child weight	0.12

**Table 4 sensors-23-02320-t004:** Comparison of diagnosis accuracy.

Methods	Accuracy (%)	Methods	Accuracy (%)
SSAE-XGBoost	96.30	PE-SSAE-XGBoost	99.27
CNN	96.82	PE-CNN	97.53
SVM	94.78	PE-SVM	95.26
DBN	93.19	PE-DBN	95.67

**Table 5 sensors-23-02320-t005:** Comparison of diagnosis accuracy with unknown faults.

Methods	Accuracy (%)	Methods	Accuracy (%)
SSAE-XGBoost	86.96	PE-SSAE-XGBoost	92.34
CNN	84.19	PE-CNN	90.27
SVM	82.42	PE-SVM	89.51
DBN	81.03	PE-DBN	89.21

## Data Availability

The data sets used and/or analyzed during the current study are available from the corresponding author on reasonable request.

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
