# Peer review of "Fault Diagnosis of Rolling Bearing Based on a Priority Elimination Method"

_sensors, 2023, doi:10.3390/s23042320_

Round 1
Reviewer 1 Report
This paper proposed a priority elimination method, which combines with SSAE and XGBoost to improve the fault diagnosis accuracy of rolling bearings and the identification ability of unknown faults, and the CWRU experimental studies seem to show the effectiveness of the proposed method. However, there are many researches on this subject, this paper needs to be improved before publication.
(1) The abstract needs to be simplified, it is not necessary to list the recognition accuracy of CNN, SVM, and DBN.
(2) The following related published paper might be helpful for the literaturereview of condition monitoring based on deep learning: 10.1016/j.measurement.2022.111997, 10.1016/j.apacoust.2023.109225, 10.1504/IJHM.2021.118009.
(3) As shown in Table 3, Parameter setting of SSAE-XGBoost are shown. How are these parameters determined?
(4) The proposed method is compared with classical fault diagnosis methods, such as CNN, SVM, and DBN. However, ther are no iinformation about the parameter setting of these models. In addition, as the input of CNN and DBN are 2-D data and that of SVM is 1-D data, what are the input of these models?
(5) It is suggested to add the comparison between the proposed method and PE-SSAE, PE-CNN, PE-SVM, PE-DBN.
Reviewer 2 Report
The authors are proposing a fault diagnosis method based on the Priority Elimination Method. I believe the proposal is a novelty. However, the authors should emphasize this point. There are some observations that can improve the quality of the text. I believe that the methodology presented is confusing. In the present format and structure, the paper cannot be recommended for publication.
1) The authors comment in the third paragraph that there are two kinds of commonly used methods for feature extraction: signal processing and deep learning. In these terms, it seems that the methods are excluding. However, they can be used together. This idea should be reformulated.
2) Section 2 should be justified. I understand the theoretical bases are presented. But you do not know the reason for each method until section 3. The authors should the clear about the utility of each method.
3) I believe that the input and output of each step in the flowcharts of Figures 2 and 3 should the presented. Besides, it seems to me that the nomenclature in section 2 is confusing. Is the y of section 2.1 the same as section 2.2? The nomenclature should be defined and changed if it is necessary.
4) What is Formula (26) on page 6?
5) There are some controversies about Case Western Reverse University on literature. It is referenced and applied in many papers, but there are some that criticize it. The authors should be attentive to this fact.
6) The comparison to other methods is very important. However, it is meaningless here, because there is no information about the design, chosen parameters, and efficiency of those methods. It is important to detail how those methods were set up.
Due to these weaknesses, I do not recommend the paper for publication in its present form. I believe the authors can improve the quality of the paper, by answering these issues.
Round 2
Reviewer 1 Report
The quality of the paper has been significantly improved, and I think it can be published on Sensors.
Reviewer 2 Report
I am satisfied with the improvements. I believe the paper can be accepted for publication.